# Leptomeningeal Carcinomatosis in Early Gastric Cancer: A Case Report and Literature Review

**DOI:** 10.3390/healthcare12121184

**Published:** 2024-06-12

**Authors:** Alessio Lucarini, Giulia Arrivi, Elena Liotta, Francesco Saverio Li Causi, Leonardo Di Cicco, Federica Mazzuca, Mattia Falchetto Osti, Genoveffa Balducci, Paolo Mercantini

**Affiliations:** 1Surgical and Medical Department of Translational Medicine, Sant’Andrea Hospital, Sapienza University of Rome, Via di Grottarossa 1035, 00189 Rome, Italy; francescosaverio.licausi@uniroma1.it (F.S.L.C.); leonardodicicco1@gmail.com (L.D.C.); genoveffa.balducci@uniroma1.it (G.B.); paolo.mercantini@uniroma1.it (P.M.); 2Oncology Unit, Department of Clinical and Molecular Medicine, Sant’ Andrea University Hospital, Sapienza University of Rome, 00189 Rome, Italy; giulia.arrivi@uniroma1.it (G.A.); elena.liotta@uniroma1.it (E.L.); federica.mazzuca@uniroma1.it (F.M.); 3Department of Medicine and Surgery and Translational Medicine, Radiotherapy Oncology, Sapienza University of Rome, Sant’Andrea Hospital, 00189 Rome, Italy; mattiafalchetto.osti@uniroma1.it

**Keywords:** leptomeningeal carcinomatosis, gastric cancer, brain metastases

## Abstract

Leptomeningeal carcinomatosis (LC) is a rare site of metastasis in solid tumors, and it is associated with poor prognosis due to disabling symptoms and a scarcity of treatment options. This condition is an uncommon entity in gastric cancer (GC). We present a case of primary LC manifestation in a patient with an incidental diagnosis of localized node-negative GC. We additionally perform a literature review and discuss the diagnostic and therapeutic challenges. In conclusion, LC from GC represents a rare condition with a dramatic prognosis. Its diagnosis might be very challenging. A multidisciplinary approach appears to be the best strategy for the management of LC from GC.

## 1. Introduction

Gastric cancer (GC) is the sixth most common cancer, representing 5.6% of newly diagnosed cancers and the third largest cause of cancer-related deaths worldwide (7.7%) [1]. Its incidence and mortality are constantly decreasing over time due to screening policies and better hygienic condition worldwide [1]. Epidemiologically, GC displays a higher distribution in Asian countries [2], and some authors suggest using different therapeutical approaches between Eastern and Western patients presenting with GC [3,4,5]. Currently, the best treatment for gastric cancer amendable for resection is an interplay between perioperative chemotherapy followed by surgery [5,6,7]. For non-resectable/metastatic patients, current guidelines recommend various chemo- and immuno-therapy approaches [6,7].

The most common sites of GC metastases are represented by the locoregional and distant lymph nodes, liver, lung, and peritoneum [8]. Brain metastatic localizations of GC, associated with poor prognosis, usually involve cerebral parenchyma and are correlated with amplification of human epidermal growth factor receptor-2 (HER2), Caucasian ethnicity, proximal location, and histological signet cell ring subtype [9,10,11]. A more unusual and rarely reported site of metastatic spread is the meninges route, causing leptomeningeal carcinomatosis (LC). Very few cases in the literature describe instances where the first manifestation of GC is LC [12,13,14,15], and in only one patient was a locally advanced node negative GC observed [13]. The clinical presentation is characterized by common and non-specific neurological symptoms, including headache, nausea and vomiting, seizures, as well as isolated cranial nerve palsy. No specific therapies are available and systemic treatment has a scarce efficacy; therefore, therapeutic options mainly consist of symptom palliation with anti-depressants, anxiolytics, and opioid and non-opioid agents [16].

## 2. Materials and Methods

We present the case of 77-year-old man affected by localized gastric cancer with leptomeningeal spreading. Furthermore, a literature search through the PubMed database was conducted to identify articles regarding patients with a clinical primary manifestation of exclusive leptomeningeal carcinomatosis from gastric cancer, while patients without a positive anamnesis for gastric cancer and/or gastric resection were excluded. The search strategy was as follows: “stomach neoplasms [mesh] AND lepto* [tiab] AND carci* [tiab]”. Screening of titles, abstracts and articles in English language was performed. Articles in languages other than English were included, provided that they had an English abstract with sufficient information to extrapolate data for our review. Only studies published between 01/2000 and 02/2024 were included in our review.

## 3. Results

### 3.1. Case Presentation: Clinical History and Findings

A 77-year-old Caucasian male referred to the emergency department (ED) of our hospital in December 2023 for worsening asthenia and cachexia over the previous two weeks, presenting paradoxical dysphagia and a weight loss of almost 20 kg in the last year (−30% of body weight). On his previous medical history, the patient had a systemic hypertension under medical treatment, and in 2020 he developed a left vocal cord palsy associated with dysphonia. The neurological exams revealed spontaneous fluent speech, appropriate in form and content, preserved extrinsic ocular motility without nystagmus, hyposthenia of the left superior and inferior facial muscles with left Bell phenomenon, ipsilateral hearing loss, and protruded tongue as a muscle deficit. Upon blood exam, the patient showed only mild anemia (hemoglobin 12.6 g/dL, red blood cell 4.2 × 10^6^/μL). The patient was then admitted to the neurology department and a brain and encephalic trunk magnetic resonance (MRI) was requested. The patient’s MRI results showed a thickening of the intracisternal tract of the V pair of cranial nerves, with hypersignal in T2-FLAIR sequences and contrast enhancement. The same characteristic was also documented for the VII, VIII, IX, X, and XI pairs of cranial nerves. A linear enhancement of the intracisternal tract of the III and VI cranial nerves was noted. Also, an enhancement of the leptomeningeal surface of the spinal cord was documented in C1 and C2 (Figure 1).

This finding was depicted as non-specific, and the differential diagnosis included inflammatory, neoplastic, neurolymphomatosis, or granulomatous disease. During hospitalization, the patient was tested for the extractable nuclear antigen antibodies (ENAs) profile (Anti-Ro52 (Sjögren Syndrome A, SSA), anti-Ro60 (SSA), anti-La (SS-B), anti-Smith (Sm), anti-Jo1, anti-systemic sceloris (Scl 70), anti-ribonucleoprotein (RNP), cytosplamatic-anti-neutrophil cytoplasmatic antibodies (cANCA), perinuclear ANCA (pANCA), anti-nucleus antibodies (ANAs)), the paraneoplastic antibody panel, and immunoglobulin G subclasses; all results were negative. Afterwards, the patient was scheduled for a lumbar MRI that showed pseudonodular enhancement after contrast injection of the medullaris conus and the origins of the cauda equina (Figure 2). A neurolymphomatosis was then suspected by the radiologist, rather than a neoplastic leptomeningitis, and for this reason a lumbar puncture was performed, showing a xanthocromic clear fluid with an elevated cell count (49 mm^3^, 70% lymphocytes, with some bigger lymphocytes containing nuclear abnormalities, 87% T lymphocyte with a T4/T8 ration of 5.2, 2% B lymphocyte, 12% natural killer), hypoglycorrhachia (34 mg/dL) and hyperproteinorrachia (350 mg/dL), negative for bacterial growth, negative for viral infections (herpes simplex virus-1, herpes simplex virus-2, cytomegalovirus, varicella zoster virus, Epstein–Barr virus, and human herpes virus 6). The cytologic exam of the liquor showed erythrocytes, rare neutrophilic granulocytes, and occasional histiocytes, with a slight increase in the lymphocyte quota noted.

A full-body computed tomography (CT) scan with intravenous contrast showed a pre-pyloric contrast-enhanced thickening (9 mm) in the stomach without any locoregional lymphadenopathy (Figure 3).

To further investigate the gastric findings, the patients underwent a gastroscopy that confirmed a pre-pyloric lesion with a positive biopsy for signet ring cell gastric adenocarcinoma with microsatellite stability (MSS) and immunohistochemistry 1+ for HER2. An in-depth anamnestic investigation revealed no upper gastrointestinal tract symptoms, despite progressive and disabling neurological symptoms.

In the meantime, after a neurosurgery consultation, the patient was proposed for a biopsy of the causa equina, which showed metastatic spread of a signet ring cell gastric adenocarcinoma c-erb2/NEU 2+ (antibody c-ErbB2 DAKO); negative fluorescence in situ hybridization (FISH) was utilized. The mismatch repair (MMR) panel showed nuclear expression of the protein for human mutL homolog 1, Postmeiotic Segregation Increased 2, and MutS homologs 2 and 6 (hMLH1, PMS2, hMSH2 and hMSH6), demonstrating a phenotype with microsatellite stability (MSS).

During hospitalization, the patient’s neurological conditions worsened, with progressive dysarthria and cranial nerves deficit.

Due to the complexity of the clinical case and the need for several professional’s expertise, the gastrointestinal multidisciplinary team discussed the patient. Given their symptoms, comorbidities, and performance status PS (Karnofsky score 40%) a chemotherapy regimen was not deemed feasible, and whole-brain radiation therapy followed by best supportive care was suggested for the sole purpose of symptoms palliation, according to the current guidelines of the European Association of Neuro-Oncology (EANO) [17].

### 3.2. Literature Review

A total of 36 articles were found with the aforementioned search criteria; after title and abstract screening, 9 records were excluded for irrelevant topic, while were excluded 15 for a known positive anamnesis for operated or palliated gastric cancer. A total of 12 papers, all case reports, were included in our final analysis [10,11,12,13,15,16,17,18,19,20,21,22]. A total of 12 patients were retrieved from our analysis, 6 males and 5 females, and in one case the gender was not specified; the median age was 58.9 years (range 40–80). The histology reported a signet ring cell carcinoma and poorly differentiated gastric carcinoma in eight (67%) and four cases (33%), respectively; HER-2 status was investigated only in one case, and it was found to be negative [18]. The median overall survival from diagnosis, available for only eight cases, was 60 days. All the characteristics are listed in Table 1.

## 4. Discussion

Due to its clinical presentation and diagnostic difficulty, the incidence of meningeal carcinomatosis and its prevalence among cancer patients is uncertain, and the best estimate is that it occurs in 2% to up to 10% of patients affected by malignancies during the disease’s course [19].

Although frequent in leukemias and lymphomas, it represents a rare entity in solid tumors [30] and often occurs following the involvement of other parenchymal organs [20]. Lung (highest absolute number), breast (highest probability), and melanoma solid tumours develop meningeal carcinomatosis most frequently. In breast cancer patients, meningeal carcinomatosis is most commonly associated with young age, ductal carcinoma, HER-2-positive tumours, and triple-negative tumours. In lung cancer, meningeal carcinomatosis is frequently associated with adenocarcinoma histotype, oncogenic driver mutations like epidermal growth factor receptor (EGFR) mutation, and anaplastic lymphoma kinase (ALK) translocation. In the largest cohort of patients with melanoma, BRAF mutations were identified in 67% of patients with leptomeningeal carcinomatosis versus 47% in the general population of patients with melanoma [17,19,21,22]. Although a rare occurrence, meningeal carcinomatosis appears in 0.062% of patients with gastric cancer [31].

Typical signs and symptoms of LC are mainly headache, nausea and vomiting, altered mental status, and also cranial nerve palsy manifested through sensory loss or facial paralysis, as they depend on the central nervous system (CNS) area of meningeal carcinomatosis involvement. However, clinical presentation may be asymptomatic or very subtle, with minimal to no symptoms [17,30].

The largest database on LC currently available in the literature, constructed by Megid et al. [32], reports a rate of 0.61% LC over more than 3200 patients with esophago-gastric cancers. The authors report a higher rate of HER-2 positivity in brain metastases over LC from esophageal and gastric cancer. The authors concluded that the best treatment in LC appears to be whole-brain RT, but 35% of patients might be planned for best supportive care. A discrepancy between the two proposed treatments in terms of survival is clearly evident (2.8 months versus 0.7, *p* = 0.015).

According to the EANO-ESMO Clinical Practice Guidelines (2023) [17,30], leptomeningeal metastasis diagnostic work-up includes a detailed neurological examination; cerebrospinal MRI with and without contrast, as the gold standard imaging method in LC diagnosis and follow-up; and lumbar puncture, when possible. CT scan should be restricted to patients with contraindications to MRI or emergency settings. Leptomeningeal biopsies are often unnecessary but may be useful in cases of difficult differential diagnoses, such as patients without a history of cancer and negative cerebrospinal fluid (CSF) cytology [17].

Typical MRI signs of LC include “contrast enhancement of cerebellar folia and sulci, basilar cisterns, cranial nerves, brain surface, the surface of the lateral ventricles and lumbar nerve roots, notably the cauda equina” [17], or they may be completely non-specific and should be interpreted in the context of clinical signs. According to a retrospective study conducted in 2020 by Le Rhun et al. [17,33], five groups of LC cases were described in MRI with different roles of prognostic and predictive value [34,35,36,37]. Abnormalities on MRI were observed in only 67% of patients with LC; therefore, a normal MRI might not rule out LC [35,36]. Cytological examination of the CSF is currently considered the gold standard for the diagnosis of LC, but its positivity does not exceed 60%, thus a repeated lumbar puncture might be needed [17,38].

The diagnosis of LC from GC is eased by radiological findings of a positive anamnesis for gastrointestinal manifestation, and it can be suspected in advanced disease with metastatic involvement of the liver, peritoneum, or brain. It should be taken into account that in our case, the patient did not report gastrointestinal symptoms, and the CT scan showed a locally confined GC without locoregional lymphadenopathy; this indolent and subclinical presentation may have slowed the diagnostic work-up.

A multimodal approach is required for treatment options [39,40], which include intrathecal chemotherapy in association with systemic chemotherapy, and radiation therapy according to the current EANO guidelines [17]. A patient’s performance status and comorbidities, histological and molecular tumor characteristics, and previous treatments should be considered. Regarding radiotherapy, a stereotactic approach may be preferable in nodular meningeal carcinomatosis, whereas whole-brain radiotherapy therapy is preferred in extensive nodular LC, although it is not associated with improved OS [17].

In our literature review, we found that two patients underwent intrathecal methotrexate, two patients underwent whole-brain radiation therapy, and only one underwent systemic treatment with the FLOT protocol [41], while most patients were assigned to best supportive care due to their associated poor performance status. In our case, the patient was proposed for whole-brain radiation therapy to palliate the neurological symptomatology, and no other treatment was proposed considering his performance status.

## 5. Conclusions

In conclusion, LC from GC represents a rare condition with a dramatic prognosis (usually less than 2 months). Diagnosis might be very challenging, and the available treatments are poorly effective due to the disease’s anatomical location and very rapid clinical worsening. A multidisciplinary approach with a focus on supportive care appears to be the best strategy for the management of LC.

## Figures and Tables

**Figure 1 healthcare-12-01184-f001:**
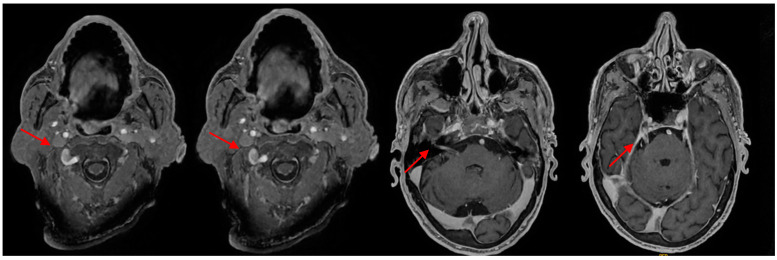
Brain MRI.

**Figure 2 healthcare-12-01184-f002:**
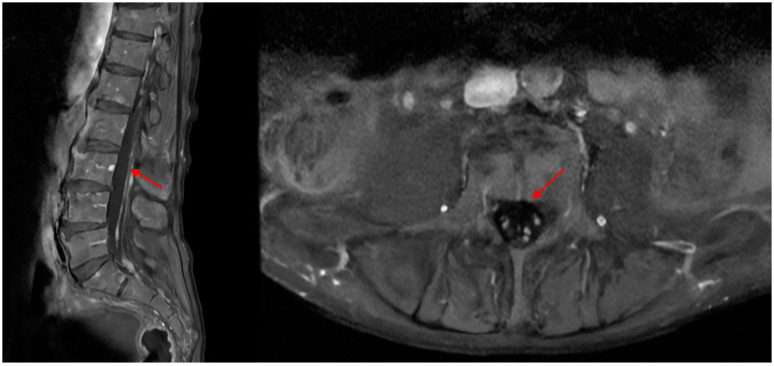
Lumbar MRI.

**Figure 3 healthcare-12-01184-f003:**
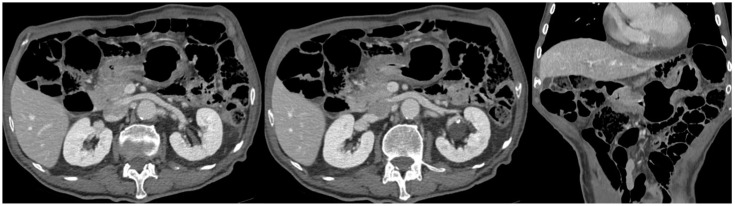
Abdominal CT.

**Table 1 healthcare-12-01184-t001:** Literature review of cases with initial manifestation of meningeal carcinomatosis from gastric cancer. NR: not reported, RT: radiation therapy, S1: tegafur/gimeracil/oteracil, FLOT: fluorouracil, leucovorin, oxaliplatin, and docetaxel.

Study	Country and Year	Age and Sex	Clinical Presentation	Pathology	Treatment	Outcome
Fuchizaki U et al. [23]	Japan, 2005	42, male	Unsteady gait, ataxia, dysmetria	Signet ring cell carcinoma	Chemotherapy	Deceased 49 days after diagnosis
Braeuninger S et al. [24]	Germany, 2005	68, male	Vertigo, diplopia, nausea, vomiting	Poorly differentiated carcinoma	Intrathecal methotrexate	Deceased 2 months after diagnosis
Lee G H et al. [25]	South Korea, 2007	49, female	Headache, dizziness, easy fatigability, and melena	Signet ring cell carcinoma	Supportive care	NR
Cresto N et al. [26]	Switzerland, 2007	57, NR	Nocturnal limb hypoesthesia and paresthesia, visual impairment	Signet ring cell carcinoma	NR	NR
Suto C et al. [27]	Japan, 2007	70, male	Optic neuropathy	Signet ring cell carcinoma	NR	Deceased 3 months after diagnosis
Yamada T et al. [28]	Japan, 2008	53, male	Anorexia, intermitting diplopia, general fatigue, headache, vertigo	Poorly differentiated carcinoma	Whole-brain RT	Deceased 127 days after diagnosis
Ohno T et al. [14]	Japan, 2010	62, male	Bilateral hearing loss	Poorly differentiated carcinoma	Whole-brain RT + S1 + Paclitaxel	Deceased 12 weeks after diagnosis
Kawasaki A et al. [12]	Japan, 2014	80, female	Headache, nausea, fever	Signet ring cell carcinoma	Supportive care	NR
Guo J-W et al. [13]	China, 2014	40, female	Headache and cervical pain	Signet ring cell carcinoma	Supportive care	Deceased 2 months after diagnosis
Vergoulidou M [15]	Germany, 2017	48, female	Headache and nausea	Signet ring cell carcinoma	Intrathecal methotrexate + systemic FLOT	Deceased 2 months after diagnosis
Ino R et al. [18]	Japan, 2021	77, female	General malaise, posterior neck pain, gait disturbance	Poorly differentiated carcinoma	NR	Deceased 25 days after diagnosis
Silverman A et al. [29]	USA, 2023	61, male	Positional headache, blurry vision, early satiety, weight loss	Signet ring cell carcinoma	NR	NR

## Data Availability

Data are contained within the article.

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
