# Peer review of "Leptomeningeal Carcinomatosis in Early Gastric Cancer: A Case Report and Literature Review"

_healthcare, 2024, doi:10.3390/healthcare12121184_

Round 1

Reviewer 1 Report

Comments and Suggestions for Authors

Lucarini et al present the case of a 77 year old male with localized gastric cancer with signet ring features who developed symptomatic leptomeningeal metastatic disease and no other sites of disease. This is a rare phenomenon that is very rarely seen in practice and accurately diagnosed. A few recommendations:

1. I think the language can be improved to make the manuscript easier to read. For example, line 30-31 “ authors suggest a different therapeutical approach between eastern and western tumor (3–5).”, line93-94 “The cytological exam of the liquor showed on the blood background..”

2. Figures 1-2. Recommend the authors use arrows to point at the abnormal areas. 

3. Image 2 and 3 are under the same “Figure 2” name.

4. Recommend the authors expand their literature review to include one more database (such as EMBASE for example) to identify literature that can enhance the discussion of their manuscript, such as Megid et al, J of neuro-oncology, 2024 PMID 38372902.

Comments on the Quality of English Language

Overall, good quality. Minor editing needed.

Author Response

Dear reviewer,

Thank you very much for your welcomed and very appreciated comments.

1, 2, 3. In the manuscript you will find the correction.
4. The discussion has been implemented with the article you cited.

Reviewer 2 Report

Comments and Suggestions for Authors

In the article entitled “Leptomeningeal carcinomatosis in early gastric cancer: a case-report and literature review” by Lucarini et al., the authors presented a case of primary manifestation of leptomeningeal carcinomatosis (LC) in a 77-year-old patient with an incidental diagnosis of localized node-negative gastric cancer (GC). The authors described the clinical history and results of neurological exams, blood exams, brain and lumbar MRI, CT scan etc. Furthermore, the authors conducted a literature review and summarized their findings regarding primary clinical manifestations of LC in GC patients. Overall, I enjoyed reading the article and believe that this article will improve our current understanding of LC and GC pathophysiology.

Specific comments:

1.       About 2/3rd of the patients (8 out of 12) whose case is summarized in Table 1 are from eastern Asia (6 from Japan, 1 from China and 1 from South Korea). Does the ethnic background play a role in the disease pathology?

2.       Table 1. Two patients who underwent whole brain radiation therapy were quite younger (aged 53 and 62) than the patient whose case report was presented (aged 77). How would the authors explain/justify this?

Author Response

Dear reviewer,

Thank you very much for your welcomed and very appreciated comments.

  1. The ethnic background might play a role (different biology, risk factors etc.), but to the best of our knowledge we currently don't know if leptomeningeal carcinomatosis develops more frequently in a subset of patients more than the others. Probably (and much simpler) the higher incidence of gastric cancer in eastern countries might be the easier answer.
  2. We actually don't know why the other 2 patients underwent RT. Our patients was admitted to the ward almost exclusively for neurological symptoms and very poor performance status. Therefore we proposed the patient for RT in order to improve his symptoms without affecting his performance status. Probably this could be the same reason for the other patients.

Reviewer 3 Report

Comments and Suggestions for Authors

In this manuscript entitled "Leptomeningeal carcinomatosis in early gastric cancer: a case- report and literature review", the authors report an interesting case of Leptomeningeal carcinomatosis in early gastric cancer. According to existing research reports, this is a very rare and uncommon case. I think the manuscript is well organized, but there are still the following issues:

1. The patient's pathological information provided by the author seems insufficient. If possible, more pathological information should be provided, such as Microsatellite situation. As a rare case, providing more pathological information can provide a very effective reference.

2. This path of metastasis seems completely unreasonable. I personally express my great interest in this. This is definitely an interesting case.

In summary, the author's manuscript can be accepted after revision

Author Response

Dear reviewer,

Thank you very much for your welcomed and very appreciated comments.

1, 2. We provided an integration of molecular features in the manuscript.

Reviewer 4 Report

Comments and Suggestions for Authors

An interesting case study with potential useful to other clinicians since it is a rare cancer. However, the following concerns should be addressed to make the manuscript better, before it can be considered for publication:

1.         Abstract -  It would be better if the authors include a conclusion in the abstract.

2.         Introduction, paragraph 1 – Would be good to highlight the issue with current diagnosis and management of GC. The current paragraph may have underestimated the issue of GC since authors indicate “Its incidence and mortality are constantly decreasing over time due to screening policies and better hygienic condition worldwide” 

3.         Introduction, paragraph 2 – Would be good to highlight why LC is very rare? Is it a limitation in diagnosis or generally it is not a site that tumours will form/metastasis ?

4.         Method : Why authors only include “Only studies published between 54 01/2000 and 02/2024 were included in our review” ?

5.         Section 2.1 maybe more appropriate to be classified under “Results”

6.         Line 57 : Should include the ethnicity of the patient since it was mentioned in the Introduction, whereby Asian seems to have a higher prevalence. Is this patient from which Asian origin?

7.         Line 60 “patient has a systemic hypertension under 60 medical treatment” - Would be good to include the detailed drug history of this patient since would this be a medical induced phenomena/side effect? 

8.         Line 65 :  How long has the patient suffers anemia ? Any sign or symptoms of anemia from patient’s past medical history?

9.         Line 68 and Figure 1 “On MRI the patient showed a thickening of the intracisternal tract of the V pair  of cranial nerve with hypersignal in T2-FLAIR sequences and contrast enhancement” Would be good to clearly indicate with arrows in the MRI (figure 1), where is the above observation noticed.

10.   Figure 1 title should be more descriptive of the key findings from the Figure 1, rather than just “Brain MRI”,

11.   Line 85 and figure 2 “pseudonodular enhancement after contrast injection of the medullaris conus and the origins of the cauda equina (Figure 2)” - Would be good to clearly indicate with arrows in figure 2, where is the above observation noticed.

12.   Figure 2 title should be more descriptive of the key findings from the Figure 2

13.   Overlapping numerical numbering of figures : Figure 2 Lumbar MRI and Figure 2 Abdominal CT

14.   Line 110, define abbreviation at first use, example : NEU, ErbB2, DAKO

15.   Line 114-118 – The clinical decision by the healthcare team is it well supported by any literature/guideline? If yes, would be good to include here. If it is not, authors may need to provide more justification on why the team has decided this clinical approach rather than other clinical approaches.

16.   Line 117, does whole-brain radiation therapy was given to the patient? The description seems confusing and perhaps authors may want to clarify it.

17.   Table 1 – the specific chemotherapy agent and its route of administration for “Fuchizaki U et al” should be specified, just like other studies in Table 1.

18.   Table 1 – What is S1 ?

19.   Table 1 – What is FLOT?

20.   Line 168 – Is it a reason why Le Rhun et al. study is not included in Table 1?

21.   Line 169 – Do confirm whether any typo-error in “value e different roles”

22.   Line 181 – “intrathecal chemotherapy in association with systemic chemotherapy, and radiation therapy” Whats are the specific chemotherapy agent ?

23.   Line 187-192 – A little more detailed description is needed as currently description is insufficient to support clinicians that may encounter the same case in the future.

24.   Line 194 – Whats the meaning of “dramatic prognosis”? Authors may need to be more specific.

25.   Line 196 – Why the available treatment is poorly effective? Is it due to delivery limitation, the drug’s potency or the drug resistance issue ?

26.   Line 197 – Is supportive care the only option for the patient?  There is no other experimental drug or therapeutic approach ?

27.   Learning from this case study, how may we improve the management of LC if there is a new case of LC diagnosed by other team? Authors may want to include a more constructive remark for other clinicians.

28.   Line 205 – Is it patient or patients  ?

29.   Authors may want to address the inconsistent references format under “Reference”.

Comments on the Quality of English Language

Minor grammatical and typo-errors. Authors should proof read the manuscript carefully.

Author Response

Dear reviewer,

Thank you very much for your welcomed and very appreciated comments.

  1. It has been implemented in the manuscript;
  2. It has been implemented in the manuscript;
  3. The leptomeningeal route is a very unusual localization of metastasis from solid tumours, we will implement this in the introduction;
  4. We used the 2000s cut-off to set a time limit. We performed the literature search at the beginning of 03/2024;
  5. We modified it in the manuscript;
  6. The patients is caucasian, we implemented it in the manuscript;
  7. It is very unlikely that anti-HT might play a role in metastatic spread. There is no evidence to the best of our knowledge;
  8. Sadly, we have no data of previous blood exams;
  9. It has been implemented in the manuscript;
  10. It has been implemented in the manuscript;
  11. It has been implemented in the manuscript;
  12. It has been implemented in the manuscript;
  13. In the manuscript they are referenced correctly;
  14. Those mentioned are not considered to be acronym;
  15. It has been implemented in the manuscript;
  16. It has been clarified in the manuscript;
  17. The paper did not provide any additional information on the type of chemotherapy;
  18. It has been clarified in the table;
  19. It has been clarified in the table;
  20. The paper from Le Rhun et al. does not meet our inclusion criteria;
  21. Typo, fixed;
  22. The chemotherapy regimen are the standard regimens recommended by the guidelines on gastric cancer;
  23. It is stated in the paper that the therapeutical approach we chose was required due to the poor performance status and neurological symptoms;
  24. It is stated in the paper that the prognosis is usually less than 2 months;
  25. The treatment is considered poorly effective due to the very uncommon scenario, and the subsequent poor knowledge on this topic;
  26. Supportive care was the treatment we chose for our patients (see point 23), the standard treatment is recommended by the guidelines cited in our work;
  27. As we mentioned, multidisciplinary approach is the best choice for this kind of clinical scenario;
  28. Fixed;
  29. What are the inconsistent references?